# Reflections for a Sociological Representation of the Eater

**Jean Pierre Corbeau**

IEHCA, Université de Tours, 37000 Tours, France; jean.corbeau@univ-tours.fr

**Abstract:** Professor Jean Pierre Corbeau is an important author of the French sociology of food. He played a decisive role in the emergence of the concept of the eater. This essay is a reflexive discussion by the author of one of his theoretical articles published in 1997. It is an opportunity for the English-speaking sociological community to become better acquainted with this current in the sociology of food.

**Keywords:** sociology of food; eater; gastrolastress; ethos; interactionism

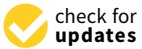



## 1. Introduction

In the late 1990s, we proposed a plural and situational approach to eating (Corbeau 1997a). Today, we want to return to this text, not to disavow it, but on the contrary to consider it as data, as a reference point—already inherited from previous studies and methodologies—from which a critical reflection (including both the temporal mutations of the observed behaviors of the eaters and the ways of reporting them, of representing them) is pointing to the dynamics of the actors, new conceptions of a typological approach, and is helping to update our sociological representation of the eaters.

After having specified our theoretical references and their methodological consequences, we will define a form of dynamic and situational representation of the eater; what we call the "ethos". This "utopian typology" is built by combining the notions of the channel and eating triangle. We will, more particularly, materialize this representation, which results from an empirical approach, through the example of the "gastrolastress" (emergence of an intermittent behavioral model), the true template of the "hopping-eaters" who reflect "hyper-modernity".

## 2. Theoretical "Filiations" and Their Methodological Consequences

The different sociological schools of thought in which we are involved must be defined in order to understand the social actor, in this case the eater. They imply preferences for an observation or data collection tool. They also determine the proposed representations of eaters and their modes of consumption. We place ourselves in a perspective described by Gurvitch (1953) as "dialectical hyper-empiricism".

Our dialectical approach denies any dogmatism, escapes any form of influence from preconceived philosophical and/or conceptual views.

The choice of empiricism is justified by our intention to account for the human dimension of any action, of any social behavior. Dialectics and empiricism are combined in a methodological system that distinguishes several levels of perception (deconstructive and constructive, moving from a macro dimension to a micro-sociological dimension where boundaries are not always set) of the "total social phenomenon" (Mauss 1980) that the act of "eating" represents for us. These levels, or stages, structure and organize the emergence of the "social phenomenon", interweaving and hierarchizing themselves in a dialectical way through time and space.

"Dialectical hyper-empiricism" also requires a "sociological imagination" (Mills 1967). It implies constant critical questioning of the means used in the field to obtain information, as well as the need to constantly elaborate a conceptual model that can capture and explain

social movement and invention, as Duvignaud (1969, 1970) has shown and taught us. To practice this "dialectical hyper-empiricism", we borrow methods and points of view from three sociological schools: comprehensive sociology, constructivist sociology, and interactionist sociology.

Let us briefly recall the main borrowings from these three schools. Comprehension is an approach that draws a specific meaning from cultural phenomena and allows us to study human behaviors by attaching them to the values that guide and express them (values that are themselves produced by humans). An initial intuitive understanding of a particular phenomenon provides a plausible explanation. To move on to a valid scientific explanation, a conceptual "ideal-type" must be constructed that conceptualizes the relations between the rational aspects that are directly understandable and the non-rational aspects. The meaning thus understood about the phenomena of cultural behavior are not those of the experience lived by the subjects, but those which are reconstructed by the researcher, mediated in this "sur-real", utopian concept, which is the "ideal-type". This method is a hermeneutic of cultural meanings.

This comprehensive perspective developed by Weber (1964) is taken up by Alfred Schutz who includes it in "constructivism":

> *The objects of thought as constructed by social scientists, are based on the objects of thought constructed by the common thought of man leading his daily life among his fellow men and referring to them. In this way, the constructs used by the social scientist are, so to speak, second-degree constructs, namely constructs of constructs built by the actors on the social scene whose behaviors are observed by the scientist who tries to explain them while respecting the procedural rules of his science.* (Schütz 1962)

The "constructivist interactionism" of Peter Berger and Thomas Luckmann is more appealing to us. These authors distinguish society as an objective reality and a subjective reality. Objective reality is "externalized" by emancipating itself from the actors who produce it. It is also "objectified" (where worlds of objects are separated from the subjects) and then it feeds the process of institutionalization. Subjective reality is "internalized" through socialization.

These theoretical collusions have methodological consequences. First of all, the meaning that the actor gives to his own behaviors is usually sought through the collection of life stories or narratives that allow us to grasp the trajectories and social logics part of an approach that is qualitative. This approach gives an important role to symbols and imagination. Our comprehensive sociology, which seeks to produce meaning from the information collected in the field[1], is necessarily phenomenological and has a double dimension: synchronic and diachronic. An ethno-methodological observation then allows us to collect data on forms of sociability, rituals, and food sharing in various situations (this approach was recently taken up by Héron 2016). This observation, set in the perspective of the restricted groups or individuals who compose them, is necessarily completed by a macro-sociological knowledge of the dominant trends of the societies to which these actors belong. We also take into consideration the media discourses likely to play the role of mentors in our societies[2].

## 3. The Conception of "Ethos"

Our "interactionist" logic makes us distinguish "sociality" from "sociability" (Corbeau 1991a, 1997b). These concepts are often perceived and mistaken in a metonymic approach, but we give them different meanings. Applied to individuals, "sociality" represents their status as produced culturally by distinctive forms of socialization, a status that inserts them into multiple trajectories likely to be objectified; a status that entangles them in hierarchies and orders that sometimes may act as true determinisms. We regard "sociality" as the crystallized impact on individuals of cultural models that prevail in a given worldview. This worldview decides, within a culture[3], what must be acquired by its members according to the place they occupy, for a given cohort, in a situation carrying economic and social hierarchies, social relations of sex, access to knowledge and know-how modes.

"Sociality" is distinguished from culture in that it is only its product, the crystallization of its representation of the status of an individual at a given moment. "Sociality" thus meets the notion of externalized objective reality as defined by Berger and Luckmann (1966). Our hyper-empirical dialectical approach sets sociality upstream from the possible meaning that a social actor may give to his culture[4]. It is an accumulation of materials made significant by the inheritance of the past and awaiting significance; this distance us here from a purely structuralist conception. This approach considers the "variant values" (Kluckhohn and Strodtbeck 1961) as potential elements of a "basic personality" (Kardiner 1939), of an "ethos" produced partly by the hierarchical construction that an actor decides upon in given trajectories and situations.

If we were to use a metaphor, we would speak of "sociality" as a tattoo, a marker that is accepted, valued, sublimated, repressed, hidden, or denied, but which one can never get rid of. This social similarity, sometimes claimed, sometimes concealed, concerns not only individuals, but objects such as products (food), thoughts, and symbolic works, seen, outside their genesis, as social products.

Finally, it should be emphasized that "sociality" should not be confused with social bonding. We understand it as solidarity with a group or a cultural system for which one asserts one's belonging or filiation through a meaningful action. "Sociality" reduces individuals to a few identical determinants, which can be combined with each other to propose templates of social identity. Social bonding, on the other hand, triggers the rituals of a claimed belonging within which the actor, as he is being reassured, can assert differences (Neuburger 1986; Corbeau 1992).

Furthermore, "sociability" is an interactive process in which individuals choose the forms of communication and exchange that connect them to others. They can then either express a will of social replication by accepting to be a simple object or product of "sociality", or develop creative dynamics through interrelations that they seek to provoke. In the first case, their behavior results only from the social factors claiming to determine it. Striving for a reassuring conformism, they adopt the social frameworks representing the precarious balance of an established or widely accepted order and the ritualized rules governing its functioning (Beardworth 1995, Beardworth and Keil 1997).

In the second case—which is where "sociability" really contributes to the emergence of a behavior—they devise strategies to satisfy their passions and desires, to invent new forms of relationship with others, and to transgress rules that are seen as unsatisfactory or outdated. Through a series of coincidences, they may also be involved in relationships that escape all predictive logic. The dynamics of interaction then give rise to the emergence of new forms of "sociability", or at least produce new meanings to the behavior, to the ritual of a consumption ceremonial.

To represent the behavioral plurality of eaters, we propose the construction of "ethos". It is a shifting form, resulting from the encounter between "centrifugal"[5] forces—drives, passions, imagination, and invention produced by ego interactions—and "centripetal" forces—civility, normalization of body images, dietary, economic, or commercial constraints, sociality, etc. This form, which corresponds to the subject's "tinkering" with these forces of different natures, thus giving meaning to his life by inventing original trajectories, is brought together, compared, and mixed to other approaches by the socio-analyst involved in a comprehension approach so that the "ethos", which is always significant, may become a representative type. The "ethos" thus becomes the catalyst for all the elements collected during the investigations. It is a pedagogical—even literary—metaphor able to give an account of the behavioral plurality of eaters in the most dynamic and totalizing way possible.



## 4. Some "Ethos" of Behavior towards Fat-Containing Product

The example of the consumption of fatty products will give a concrete expression to our statements.

Within the types elaborated from our observations in the field, our conception of "ethos" has sharpened our knowledge and understanding of the eater (Corbeau 1991a, 1991b, 1996). This conception of "ethos" distinguishes sub-segments within these major trends by taking into consideration consumption scenarios within given situations. It brings the temporal dimension into the social trajectories of eating behavior. In line with the "hyper-empiricism" that we claim, it breaks through the inertia of mere typology by pointing to the plasticity and behavioral dynamism of eaters whom we have been calling "hopping-eaters" for some years.

### 4.1. Four Main Types of Eaters

In the 1980s, we identified and constructed four main types of eaters: the "complexed overeaters", the "advocates of light nutrition", the "advocates of substantial nutrition" and the "gastrolastress". Each of them would constitute a template in which a plurality of "ethos" could be identified.

After briefly mentioning what these types covered, we shall describe how we obtained and categorized the information in order to build the "ethos". We shall then materialize some of them by considering their contemporary mutation.

The "complexed overeaters" were and still are a model of eaters that is difficult to identify based on objective food consumption. They are overwhelmed by a feeling of guilt as soon as they eat. This can be constant for some or more fleeting for others, for example, at the end of a meal, when they fear digestive disorders or start to feel the painful effects, or when they are trying on swimsuits and are struck by anxiety. In designing these four types of eaters, we identified six "ethos" trajectories within the "complexed overeaters". All of them expressed a rather simplistic and mechanical vision of a body perceived as a pipe in which food carrying toxins must flow as quickly as possible in order not to contaminate, pollute, or soil the body. These representations, suggested by the group of "complexed overeaters", were already dramatized by the media and regularly taken up by advertising and commercial strategies, reinforcing the magical vision of their food and a certain anxiety that could result from it. Today, these characteristics still constitute the "subtext" of this initial type, but the intermittent dimension of the "complexed overeaters", whose behavior varies according to the situations of consumption (solitary, commensal, or convivial), must be underlined. It is also necessary to point out the systematization of an increased distrust for certain foods (e.g., of animal origin, whose production is judged catastrophic for the future of the planet, too industrialized, too hyper-processed, etc.). Let us also emphasize that this type corresponds to contemporary urban societies in which—even if the inequalities in access to food contradict certain people's optimism—abundance versus waste is dramatized; in which the ties with the world of production are distended, creating suspicion around food products and their manufacturing. This is happening at a time when contemporary urban societies call for our reflexivity, which is a necessary component of our responsibility as individuals (Ascher 2005). The specificity of "complexed overeaters" is that this "reflexivity" always leads to a disastrous vision of "ill-being", yet it could rather build a "joyful palate" that would harness the sensory apparatus to further increase the delight of eating (Corbeau 2008).

The causes of distrust among eaters are underpinned by four fears: that of lack, that of excess, that of poisoning, and that of the gaze of others and the judgment that it entails, or that we think it entails. These fears pertain to health, to appearances resulting from our embodiment to religious prohibitions, or to fears about the destruction of our planet and its ecosystems (Corbeau and Poulain 2008; Corbeau 2004, 2020; Adamiec 2016). This first type of eater—the "complexed overeater"—combines the fear of excess, the fear of poisoning and, sometimes, the fear of the gaze of others. Without dwelling on this specific type and the "ethos" that derives from it, it should be kept in mind that it was and still is the template

of a reflexive relationship between eaters and food that is embedded in "hyper-modernity" (Giddens 1991; Ascher 2005) and which, in recent years, has been referred to as "orthorexia" (Corbeau and Hanckard 2019).

The second type of eaters, described as "advocates of light nutrition", appeared in the 1970s (these eaters belong to the middle and upper classes, to the "new bourgeoisie" described by Lambert (1987)). In the late 1980s, when our typology was conceived, this type of eaters was characterized by good compliance to dietary information and by a reasonable balance between the pleasure of eating and the concern for good health. These eaters attach great importance to breakfast considered as a proper meal, appreciating diversity in food (the search for new or exotic flavors as well as the rediscovery of regional and traditional dishes), but they simplify the structure of the meal, which is progressively lightened. The symbolic value of social success associated with the consumption of meat is less important to them. There are two reasons for this:

- On the one hand, their rather well-to-do social origin (large consumptions of meat—symbolic of social success—has been acquired via one or more generations in their family trajectory, and poultry—festive or not—is preferred to red meat). These urban eaters, who work mostly in the tertiary sector, achieve social distinction through the possibility of traveling, i.e., access to ubiquity. They participate in the emergence of a vegetarian diet that favors organic products, exotic plant products, and ancient vegetables.
- On the other hand, most of them have a good level of education and have a reflexive relationship with the body (Boltanski 1968). Although they are more familiar with dietary information than much of the population, they dramatize such information less. Even when their consumption is similar to that of "the complexed overeaters", the meaning they give to it is different. They seek to reconcile the pleasure of taste with a feeling of physical well-being.

They are over-consumers of fish, cheese, and processed milk (all of which substitute easily for meat), and fresh fruit and vegetables. They undoubtedly express through their food a form of hedonism with a great emphasis on lightness, nomadism, and wanderlust.

Thirty years after this first definition (Corbeau 1997a), this second type of eaters and the "ethos" that can be constructed within it is still very much characterized by these same behaviors. To bring it up to date, we would say that it represents the template of contemporary eaters who value products with high signs of quality (certification labels, organic farming, well-known producers, etc.) in order to enjoy eating (and often cooking). These eaters (urban, but increasingly "rurban" or wanting to immerse themselves in nature) are reassured when they can easily trace the origin of the product they eat. The main evolution of this type, over the last decade, lies in a paradoxical cohabitation that confirms the importance of the behavioral intermittency of "hopping-eaters". Alongside the over-consumption of exotic products inherited from the 90s and 2000s (which continues, even if it is decreasing slightly due to eco-friendly reasons), there is an emerging popularity for local products that combine the notion of proximity to the notion of terroir, and connect networks between those that participate in the pleasure of tasting some dish or wine. The craze for appellations of origin, together with the pleasure of visiting food markets, wine shops, and direct sales at the farm, are good examples. In reinforcing the vegetable part of their diet, they are excited about heirloom or strange vegetables and about exotic fruits (which are less and less so, as their production sometimes becomes local), and this distinguishes them from other eaters. If they do express food fears (which is not necessarily the case), they are those of excess and of the judgmental gaze of others.

Today, the return of the New Age and the search for well-being and pleasure through different food strategies have reinforced this second type of eaters, and in recent years the "bobos" have become its caricature. Moreover, this type is being overexposed by media campaigns that interweave food/health/wellness/beauty/ecological transition and that it is turning into a model for actors looking for imitations or for a safe sense of belonging.

The "advocates of substantial nutrition" were the third type of eaters that we distinguished in the 1990s and 2000s. Of modest socio-cultural origin—with the exception of an elderly fraction of the bourgeoisie who perpetuated the model of the traditional bourgeoisie (Lambert 1987)—their living standards improved during the post-war boom, which resulted in a significant increase in their energy intake, particularly regarding animal proteins and lipids. In addition, their purchasing strategies, their culinary preparation (with a large emphasis on cold cuts, sauces, and abundant fat in the cooking of meats and vegetables), and the jubilation associated with the protocol preceding their festive gatherings, place them more in an "eudemonic"[6] than "hedonic"[7] logic. Being big eaters of starchy foods, cold cuts, and meats, they favor hearty food that "holds to the body" as the body is always conceived as an instrument (Boltanski 1968) that has to render through work the energy stored up at the table.

This third type of eaters still exists. It remains characterized by a working-class social background and continues to express the fear of lack. To the detriment of certain public health policies, new junk food or at least high-calorie products meet their expectations in terms of taste (fatty, sweet, salty), calories (Drewnoski and Specter 2004), symbolism (access to a form of abundance and the warm atmosphere of fairground cooking; Dumay 1969; Corbeau 2005a, 2005b, 2010), and economics. This explains a form of stigmatization of this type of eating behavior and certain "ethos" that are developed in communications against obesity. However, it should be noted that today, these "advocates of substantial nutrition" (perhaps as a result of these campaigns targeting them), are willing to lighten their diet ("one-off" and sometimes following a period of overeating) for reasons that are most often linked to a desire to change their body image rather than for health reasons (except for seniors, who are often made aware of this by their family and friends).

The fourth type that we imagined a few decades ago was the "gastrolastress"[8] type. This model of eaters, which was perceived from the variability of their behavior and not through preferences or objectifiable consumption, concerned all the previous types mentioned. Its specificity lay in the modification of the diet. During productive time, the gastrolastress type of eaters watches their body and conforms to what they consider to be good nutrition. In the evening, during the weekend or on vacation, it is quite different.

Over time, we believe that our problematization of this type of eating behavior is at the origin of what some have referred to, a good ten years later, as "flexivores" or "flexitarians" (Blatner 2008). However, it seems to us that (for reasons intertwined with the desire to feed the world in a perspective of ecological transition and a vision of public health somewhat reduced to a simplistic fight against the obesity pandemic) this new typology retains and values primarily the relationship of eaters to the vegetal environment rather than their relationship to food in general.

We were among the first to note (Corbeau 1991a), after Lambert (1987), that the vegetal part of our diet was becoming more and more attractive (especially if we consider the upper classes and gender), but this was also coupled with forms of intermittency in food preferences which led us to opt for the term "hopping-eaters" since the 2010s.

We will take up and develop this reflection in conclusion after having exposed the tools designed to grasp the plasticity of types of eating behavior across space and time, as well as the possibility for ethos identification.

### 4.2. Triangles, Channel, and Diadrama of Eating

We propose to examine some particular "ethos". The idea is to give an account of a plurality of reconstructed images of various representative and/or significant consumers, images which have been established from empirical observations and analysis of the social field on the basis of a triangular model combining, diachronically and synchronically, the eater, the lipidic product (inscribed in a channel that gives it a symbolic dimension), and the meeting point between the eater and the food (cf. Figure 1). This helps to understand the logic of consumption through time and space.

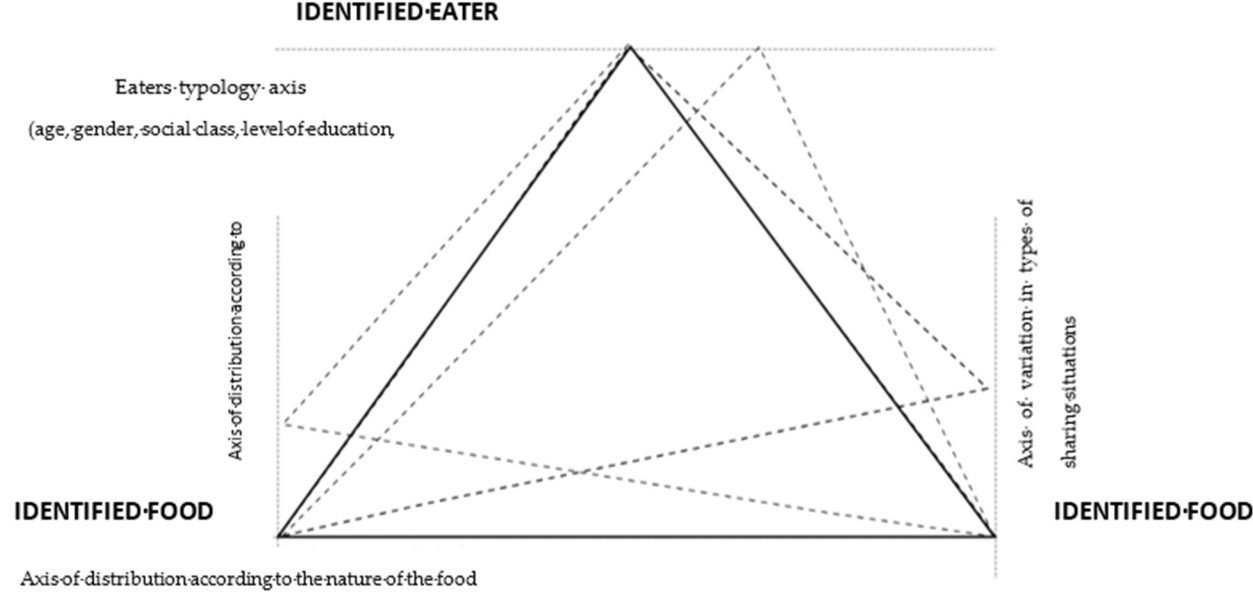

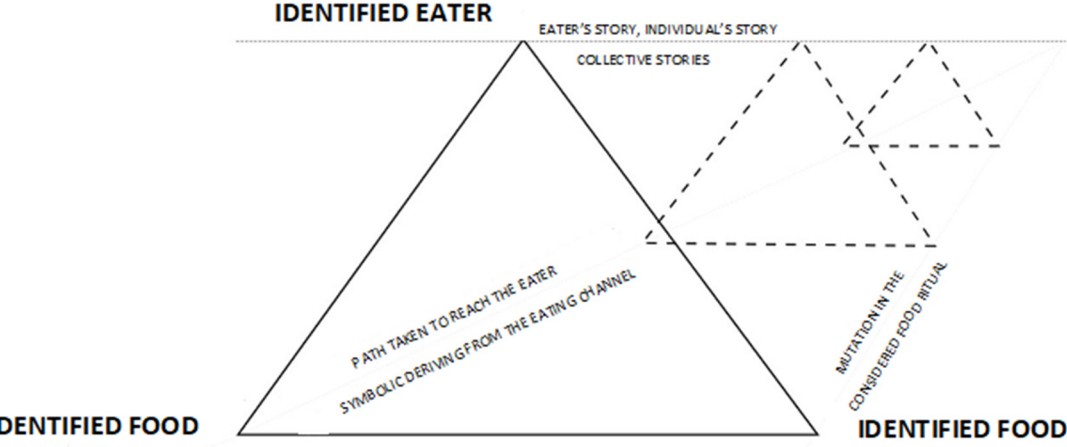

**Figure 1.** Different Triangular model combining, diachronically and synchronically, the eater, the lipidic product, and the meeting point between the eater and the food.

The "*triangle of eating*" varies in space as we consider that eaters are varied, that attitudes and behaviors change according to the individuals, but also according to the situations in which they are involved, and according to the nature of the food, its aspect, the collective imaginary which is associated with it. The" triangle" also varies in time as each of its elements has a history, individual or collective, for the eater, creating symbolism for the product (time of appearance in our societies, rareness, and journey to the eater, intersecting with the mutation of food forms, and rituals around consumption situations).

It is necessary to specify that each of the elements, synchronically and diachronically, is part of what we call the "eating channel" and the "diadrama" that characterizes this channel. These two notions help to understand the "ethos". When we refer to the concept of "eating", we are considering all the processes that allow any food, solid or liquid, to be absorbed by an eater or a drinker. Repeating the old adage, we may say that eating goes from the farmer's pitchfork to the table fork, but this would still be simplistic as, more and more frequently, before the pitchfork is stuck in the ground, "decision-makers" direct it towards the cultivation of a specific product whose production is also likely to be changed upstream by researchers (bio-technologies or others) and downstream of the fork, by memories, collective imaginary, metalanguage of the "after-eating" (that we also

integrate in the "eating channel"), their consequences on the health, the image of the body, and the social production, which partly determine our food behaviors.

The concept of "eating" thus represents the decision to cultivate a food, to produce it, to possibly process it, to distribute it (within a self-sustaining system, a short circuit or a commercial channel creating more or less prestigious symbols), to buy or to exchange it, to cook it, to prepare it, and to offer it to the eater following a culturally codified presentation and modalities of "savoir-faire"; deciding to absorb it by respecting or transgressing table norms, body schemas expressing a "sociality", by accepting symbolic and religious representations. This "eating", inscribing itself in a pattern of openness, curiosity, or obeying boundaries, leads to the constitution of metalanguage and memories likely to modify or distort the eater's habits.

Such a definition deliberately moves us away from a notion of "eating" that would be merely social replication, as the actors (decision makers, producers, processors, transporters, distributors, preparers, researchers, and eaters) are perpetually innovating, amending, and transgressing. Modernity necessarily introduces ruptures and mutations. In this definition of "eating", tradition itself no longer refers to the imprint of a past that has been passively endured, but to some meaning in which the social actor engages, to a desire to establish a filiation of models whose resurgence is potentially facilitated.

Instead of a linear and mechanical conception of the agro-food industry, which would only group together professionals according to such a scheme (Figure 2), we prefer the more dialectical conception of the "eating channel", made up of interactive zones, in which the direction of the messages can be reversed, and in which new partners of the "eating channel"—who used to escape a reduced and focused vision of the agro-industry world—are now included. We thus obtain what we call the "diadrama of eating".

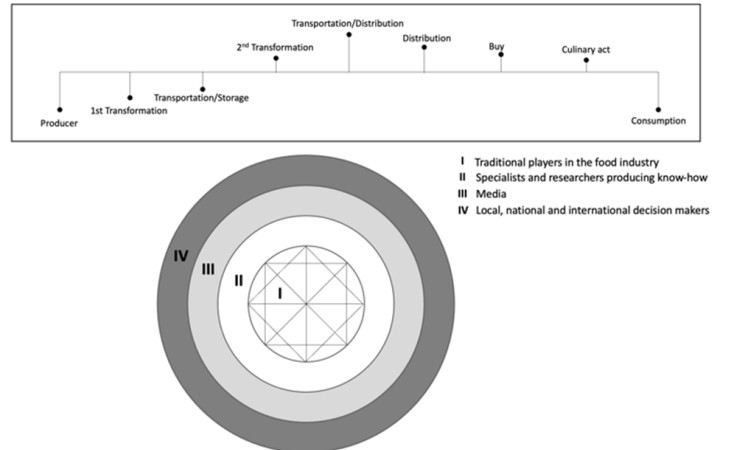

**Figure 2.** The diadrama of eating.

As for researchers, they are involved at all levels of the "eating channel" from a variety of disciplines (engineering, biology, economics, human and social sciences, medicine, ethics, etc.). It is worth noting that in the sphere of specialists observing and wishing to act in one way or another on eating behaviors, different research results are likely to be met (without questioning the quality of those who obtained them), along with interpretations of the same phenomenon or protocol that are structured according to different paradigms (hence the interest of "dialectical hyper-empiricism").

The media can take hold of these discoveries, giving them a dramatic dimension (often involuntarily, but sometimes seeking to sensationalize, by reducing and/or distorting the words of the specialist who is interviewed). At times, they favor disciplines and themes through fashionable phenomena, giving preference to certain researchers. Through a form of celebrity, these researchers are likely (no doubt unwillingly) to hide the big picture (i.e., the multiplicity of viewpoints and epistemological positions) from the eyes of the

traditional actors in the agro-industry, but also from the eyes of the decision makers (who are involved in the design of research programs and then in their financing), who are then likely to turn the more or less random choice of professional information picked by the media into a system of domination of a specific research field. If this were the case, the media power inducing all forms of decision would have adverse effects on the "eating channel".

It should also be noted that the sphere of legal, political and/or financial decision makers (local, regional, national, and international) can be in a prominent position and establish their choices as certainties that escape the other actors of the eating channel, just as the models conveyed by the media. In any case, their effects on research and education policies, the models proposed—or even "overexposed"—and the resulting consequences for our eating habits must be questioned. Finally, it should be imagined how bridges and networks can be built between the different spheres, the strategies and discourses they entail, the actions they generate, thus bringing out the "diadrama of eating" thanks to dialectical hyper-empiricism. Such a dramatized presentation of food allows us to make use of as much information as possible, multiplying the viewpoints (centripetal and centrifugal forces of the "ethos") on a possible representation of the eaters.

*4.3. Some "Ethos"*

The template of the "complexed overeaters" offers the possibility to build several trajectories leading to comparable "ethos". Some of them will be of particular interest to us.

For reasons of social appearance, the "complexed overeaters" were, until recently, essentially represented by women, as their bodies have always been more canonized, monitored, and then normalized with the advent of women's press. The "complexed overeaters" are wary of certain products and fear the magical effects of absorbing them. Since the sixties, the male body has also started being watched, and so we should discard any sexist views when observing the new concern about social appearance. In order to conform to body models, certain types of food are rejected: "fat" in all its forms, salty foods that are perceived as water-retentive, sweet foods that are imagined having a negative effect on the figure and, in general, any food that is considered to be high in calories (public health messages interact and exacerbate these pre-existing imaginary representations). Instead, the acidic flavors that we like to think of as fat-solvers are popular: lime in a fatty sauce, the sour taste of a cream that miraculously becomes light, pineapple that may also be consumed in capsules, kiwi on cold cuts. The "complexed overeaters", similar to anyone choosing certain foods for health reasons, are fascinated by diets. This is related to the magical relationship they have with food and with the conception of their body, which they compare to a tube that has to reject as much as it absorbs (and even more when it comes to being thinner or eliminating so-called toxins), or to a machine whose consumption depends on performance, and which has to be drained from time to time.

Such conceptions are also developed by the "complexed overeaters" who are overly anxious about their health. Two types of concerns arise among them: either they fear poisoning, and so they express suspicion, and even refusal, of anything they suspect is carcinogenic (chemicals from the food industry or intensive agriculture, cooked meats and their fat, etc.), or they fear cardiovascular disease (a certain form of prevention in the "diadrama" has greatly intensified this fear), which shows that they practically join the "ethos" of the "complexed overeaters" for aesthetic reasons. The only difference is that they are older, and that their magical conception of food is supplemented by protective or corrective medication.

A final "ethos" of the "complexed overeaters" puts forward ideological, philosophical, or religious arguments to refuse certain types of food. Examples include the solidarity of a certain Third Worldism during rice bowl operations, certain forms of vegetarian activism, the boycott of certain foods from countries deemed reprehensible for ideological or humanitarian reasons, and the prohibitions associated with certain major religions (Corbeau 2020). This type of "ethos" has developed considerably in recent years with the

"dramatization" of the role played by our eating habits on global warming and the strong emergence of an anti-speciesism movement condemning all forms of animal incorporation.

The "gastrolastress" type clearly reflects the variations in the collective imaginary about social times and diets. The "gastrolastress"—whose food consumption takes many forms—is characterized by a modification in their diet that depends on whether they perceive themselves in a situation of social productivity or not. Two different patterns can modify their initial food model:

In one case, sensitive to the model of the "advocates of light nutrition" valued by most of the media discourse, they will monitor their body by conforming to contemporary dietetic and aesthetic principles. To produce better, to act better, to be more efficient, they will avoid drinking alcohol, they will eat as light as possible and be wary of what seems fatty (as it could weigh down and immobilize their body, a body that boasts about ubiquity through lightness).

By rationalizing their productive time during their continuous working days, they will refuse convivial moments outside the work environment: no meals other than business meals, apart from gatherings with colleagues or nibbling in a more or less unstructured way that still enables multiple activities. Ordinary working time is opposed to the more festive time at home (when this first gastrolastress "ethos" is not solitary), during weekends or vacations. This alters both the collective imaginary around fat and eating habits. In the "gastrolastress", the time spent sacrificing productivity leads to distrusting anything designated by the media as bad for the cost of our bodies. Lipid and alcohol consumption is monitored (especially in the service professions). Should one choose cold cuts, fried food, gravy, and cheese at the company cafeteria or at a meal with colleagues, it is more by reproduction of habits established during food socialization (particularly in working-class cultural trajectories that still value the instrumental vision of the body) than by a formal desire for fat: fat is overlooked as part of the dish, it is not thought of by the eater, or alternatively, this type of eater will balance his diet by consuming green vegetables, salads, and fruit for dessert.

In the evening, back at home, spending time to enjoy life, this "gastrolastress" type might drink some alcoholic beverages, eat more fatty products than during the day (mixed finger food with drinks, casseroles, festive delicatessen, good wine, etc.). This becomes even more true over the weekend. This first gastrolastress "ethos" (of a certain socio-cultural and economic level) transgresses the cultural order that he has respected during his working time and finds a reflexive pleasure in enjoying the fat of the Sunday cake, of the country cold cuts, of the meats in sauce or confit, the unctuous cheeses, the butter or pastry creams, uncorking bottles that he will have lovingly selected in a specific distribution circuit (visit to the producer, personal cellar, wine shop, club, large and medium sized stores, etc.). These consumptions are only conceived within a sharing that creates a social bond, opposed to the individualism advocated by the technical and social division of labor.

In the second "gastrolastress" case, during productive time, one will seek calories, casseroles, or fried food that one likes to share with colleagues. "Fat" is not thought of but is present and accepted through food preferences. The point is to fill the "instrumental" body in order to exist socially through productivity. However, in the evening, in one's own territory, in a non-productive time, one will be more cautious about the "fat" eaten, especially when someone in the household is watching. One will settle for a soup, pasta, or salad with a slice of ham. The evening meal is clearly more frugal than the lunchtime meal. On weekends, unless one seeks to strengthen a social bond through traditionally high-fat family meals, one will be tempted (if one has the economic means) to experiment, as a kind of social revenge, with the "light" and exotic cuisines associated with forms of social success. One may try to replicate them using popular media recipes or, sometimes, to buy them in frozen or vacuum-packed ready-made meals whose representation matches the fashionable "lipophobia" (Fischler 1990, 1992; Poulain 2009) of the late 1980s, still prevailing in the body care within favored sociocultural categories in a productive situation.

Therefore, the construction of "ethos" integrating the notion of situation allows one to grasp a twofold relativity in the qualities of fat, bad and thought about for some, good and unthought about for others, in situations of productivity. This double representation shifts in more festive situations to good and thought about "fat" for the totality of the "gastrolastress" population, even if the biggest fans of lipids of this category will tend, during resting time, to explore a lighter diet.

## 5. Conclusions

As a conclusion to our reflections on our project to construct a sociological representation of the eater, three aspects of our approach are worth highlighting. Some of these aspects express certainties, while others are necessarily linked to more uncertain perspectives.

The certainties concern the theoretical apparatus. Of course, we will not claim to be exhaustive, yet we assert the necessity of combining the point of view of an "objectivist sociology" with that of a "comprehensive sociology". Furthermore, this twofold perspective requires the formalization of scenarios interweaving the individual (bearer of meaning) and the collective. In such an approach, modeling cannot shut us into a crystallized and reductive typology, as this typology corresponds to a precarious, if not ephemeral, balance, or to a destructuring and restructuring dynamic (Lewin 1943). We find the dialectical hyper-empiricism and its materialization through the construction of "ethos" satisfactory insofar as it enables us to escape the nominalism often associated with typologies or other mapping proposed by marketing people and based on the aggregation of a few factors identified as relevant social markers. Our methodological material, imagined from "hyper-empiricism", addresses the behavior of eaters through a complex system reconciling the rules and dynamics specific to a particular situation, with the imprint of the global society, a dimension that is often hidden yet always present.

In this perspective—and this constitutes the second aspect of our conclusions—we must question our productions, not in order to deny them, but to consider them as vectors of a dynamic construction of representations that are always evolving. In short, to grasp how the mutation of the behaviors of a "varied eater" compels us to find new ways of accounting for it, how we move from the "gastrolastress" type to the phenomena of the "hopping-eater".

As early as 1990, following on from the work of Raymond Ledrut's team, we understood the "gastrolastress" type as a distinct model of alternating eating behavior (in which the traditional social determinants allowed us to consider two different "ethos"). We were therefore reproducing, although perhaps unknowingly, a form of nominalism that we condemned in certain crystallized visions of mapping, which were certainly reassuring but simplistic. As the surveys that followed and the development of a situational approach progressed, we gradually conceived them as a new behavioral characteristic involving—to varying degrees—all contemporary urban eaters, plural eaters (Corbeau 2003) who are not just "flexivores" mutating towards a vegetarian or even "vegan diet", as recent approaches that serve both societal projects and commercial interests would have us believe. The food preferences and practices of these plural eaters, of these "hopping-eaters" (at least for those who have the economic and logistical possibility of accessing the food diversity, still revealing inequalities) are part of an intermittent mode typical of our contemporary lifestyles.

By the 2000s, our representation of "hopping-eaters" was moving away from our initial conception of "gastrolastress". They did fit into the type of plural eaters, but it was increasingly difficult to define their specific food consumption in relation to specific situations. The logic of a "transgressive indulgence" based on a preferred food model, as was the case in the first "gastrolastress" studies conducted in the late 1980s, was no longer present. Here, the eaters would imagine and experiment with various possibilities according to the situation, the resources, the available food supply, and the media dramatizations. In short, it became more and more difficult, as our surveys progressed, to characterize their

diets and preferences over a long sequence of time encompassing the multiplicity of social activities. We had to face the fact that "hopping-eaters" were finding their uniqueness in the intermittence, which created a form of individual freedom in the choice of food, its textures, the ways in which it was shared or a more individualized consumption. The "hopping-eaters" were not a new type of eater—other than the rather general type of plural eater—but an emergence of a new relationship to food and to the associated sociabilities that go along with its production and its absorption all along the eating channel. Therefore, the "ethos" is no longer constructed nor declined from a type, but rather from a posture.

In the 2020s, we analyzed this intermittency as a "total social" phenomenon interacting with forms of "anomie" (in the sense that Jean Duvignaud and ourselves give to this Durkheim concept). The "hopping-eater" of a certain age who eats, in a non-regular way, the same products as teenagers, may do it by "youthism", but he may also unknowingly express a dissatisfaction, a longing for creativity, a freedom that involves the individual and the collective. It is the same for young people claiming to cook recipes of an ancestor whom they did not necessarily know and who represents the identity of the territory of their origins to which, sometimes, they have never been, etc. Such illustrations are endless. Tradition, modernity, and their imaginary worlds are intertwined with no clear normative system prevailing. Additionally, is there even a desire? What characterizes these "hopping-eaters" is movement, a kind of trance that a stable model could never stabilize. The "hopping-eaters anomie" is a response to dissatisfaction, to a societal crisis; an uncertainty provoking a quest for new possibilities.

Only this creative anomie might weaken some of them. Without an "inner gyroscope" (Riesman et al. [1950] 2001), the "hopping-eater" becomes an "eater with fears". This phenomenon is emphasized by the jumble of injunctions of various natures (Poulain 2001; Poulain [2002] 2017). Eating becomes a dangerous game (Corbeau 2005a)! Especially when food is easily accessible, when the "gyroscope" of "the lonely man in the crowd" is gone, cracked, or broken, when mentors sometimes confront each other while hiding their true intentions (unrelated to our food), thus fueling the "conspiracy" perspective of "eaters with fears". It is obviously difficult to translate into a language other than our own (French) the play of words that we are claiming here. Yet, let us explain. In French, the concept of "hopping-eater" is "mangeur-zappeur" which sounds similar to "mangeur à peurs" meaning "eater with fears", and which also sounds similar to "mangeur-happeur" meaning "snatch-eater". This play of words illustrates how the "hopping-eater" is an "eater with fears" that becomes a "snatch-eater" of different products, in different forms of sociability, of food models or discourses that either reassure him within a logic of conformism, integrating him into reference groups—in the absence of the safe sense of belonging (Neuburger 1986)— or provide him with the possibility of tinkering, of "crossbreeding" (Corbeau 2005a) that enables him to invent or reinvent his behavior in fleeting space and time.

The second point of our conclusion also concerns the methodology. The time required to carry out such a study, which includes the diachronic dimension in its vision, may be criticized. This criticism is only justified when an initial typology of eaters is being defined for a given phenomenon (which we have done in part for the consumption of products with highly symbolic, social, economic, and nutritional stakes, e.g., fat products and wine). Then, the point is to maintain a constant observation that allows comparative analyses enabling us to grasp the dynamics of change and the complex logics of trajectories underlying food behaviors. Moreover, again with regard to the conclusions on our methodological practice, we argue for an artisanal sociology, i.e., for fieldwork, analysis and conceptualization that is not subjected to any technical (and often social) division of labor. The researcher, whatever his status, must be associated with the whole of the tasks allowing the elaboration of an "ideal-type". It is only at this price that he can understand the effect of the collective on the social actor and the meaning of the latter's response.

The last reflection of our conclusion, more prospective and less assertive, concerns our plea for a hyper-qualitative approach to the eater. This approach helps understand the upstream side of consumption, that of the values that underlie it within a consumer

mentality, a consumer who is both a replicator and a producer of norms. Our hypothesis is that a hyper-qualitative perception detects new and original markers of behavior that have not yet been acted upon. It generates congruences that make it possible to grasp the meaning of the structuring logics of the universe of eaters. As such, it is likely to be of interest to decision makers in the fields of economic, commercial, social, and health policies.

**Funding:** This research received no external funding. The cost of translation was supported by ISTHIA of Toulouse University Jean Jaures.

**Data Availability Statement:** Not applicable.

**Conflicts of Interest:** The author declares no conflict of interest.

## Notes

[1]   Here we have in mind Chebika by Jean Duvignaud, who was our mentor and accomplice, and who was one of the first authors to use a sociological method of utopian reconstruction of a totality based on facts observed in a Tunisian village from 1960 to 1965, rendered in literary form and emphasizing, through fictional characters imagined from the verbatim records and observations collected, the meaning that the actors give to their behavior, but also the meaning that the reflective distance of the researcher can attribute to them. We have in mind the studies we did with him, "La planète des jeunes", "Les tabous des Français" and with his wife Françoise, "La banque des rêves". We also have in mind our survey "Un village à l'heure de la télé", but also the work published in collaboration with Jean-Pierre Poulain, "Penser l'alimentation. Entre Imaginaire et rationalité".

[2]   Investigation methods used as part of the various studies led in collaboration with Jean Duvignaud, Jean-Pierre Poulain, Karen Montagne, Frédéric Précigout, Emilie Salvat, or by myself are semi-directive or non-directive interviews that can lead to the collection of life stories. For each study, the number of interviews I led was between 30 and 50. This enabled me to claim 'artisan' status in the field of sociology, taking part as I do in every stage of the investigation (from data collection to data analysis, taking into account the nuances of the interviewees' speech and its delivery—hesitations, intonations, mimicry, etc.—with ethos reconstruction as an outcome. The whole data collected (mine and my collaborators') could involve between 60 and 200 interviewees, depending on studies. The process of data collection stopped when items in relation to the studies' topics and the set of social criteria determined for interviewees became saturated. I added to the panel—on the advice of Jean Duvignaud—a few people identified as atypical, i.e. significant and not yet representative. I have been involved for more than 20 years, alongside my students, in this type of investigation or 'action research', enabling one, through time, to grasp mutations in eating behaviours. Finally, I have been conducting analyses of the contents of various media (press, TV, blogs) in collaboration with my students as tools to capture the inward pull interacting with the desires and preferences of eaters.

[3]   Culture is claimed here in all its semantic diversity, from the presence of elites in an imaginary museum whose access creates social distinction (Bourdieu 1979), to the anthropological conception of all the ways of living, thinking, and acting of a group at a given moment, to eventually that of a worldview marked (or reduced?) by technology or by some mode of organization or production.

[4]   Balandier (1983) published a remarkable article on the centripetal and centrifugal dynamics that shape our everyday life.

[5]   This typology finds its origin in the study carried out at the University of Toulouse le Mirail, under the direction of Ledrut et al. (1979), a study that proposed three types or major trends of food consumption patterns that we have modified and multiplied throughout our surveys since 1980 Luc Boltanski distinguishes "the instrumental representation of the body" (more present in the working classes and among men) which must be fed with a fuel necessary to the production -essential to the safeguarding of their social identity- from "the reflexive representation of the body" (more shared by the women until a recent period and typical of the privileged sociocultural categories). This reflexive conception is concerned with the consequences of the absorptions on the health and the figure and is alert to all the normative dysfunctions of the body which are then the object of care and corrections by specialists (aesthetic, medical, etc.).

[6]   That is accepting certain constraints and forms of forecasting to achieve happiness.

[7]   Unconditional searches for immediate pleasure

[8]   When I coined the word 'gastrolastress' in the 1990s (Corbeau 1991a, 1991b), I had three motives: To pay homage to François Rabelais by referencing the people of the "gastrolastress" ("gastrolaters"), whom he created as early as 1552 (in "The Fourth Book", chap. 58) as a first type of denial of the social link in the name of a preoccupation with oneself. The gastrolaters are, for Rabelais, eaters who transgress time allocated to commensality rituals when they are hungry—like gastrolastress—'making a god of their stomach', absorbing food in abundance, taking pleasure in filling their bellies and getting full. However, these "gastrolaters", like "gastrolastress", worry about their digestive health, and, while enjoying digesting when full up, fear all the ailments digestion might entail. That of "gastrolâtrie" present in Rabelais (individualism combined with the refusal to ritualize food intake by following the reactions of one's "belly"—its "needs" and its good functioning -, characteristics valued by unstructured eating) likely to take multiple forms according to different places and social times. A lineage can be established

between them and the "gastrolastress", for whom self-control in a work context relates to the concern evoked by Rabelais. This gastrolatry mocked by Rabelais can be realized in a diet reduced to its functions of nutritional contributions "good for one's health and needs", that is to say, the opposite of eating as a means of social bonding and forms of collective belonging. The second motive behind the creation of the gastrolastress is the idea of "stress" inherent to the contemporary urban actor who "rationalizes" and accelerates his productive time, is breaking with a social link of commensality and conviviality and accepting a de-structured diet perceived as accentuating individualism, a sign of social efficiency. Finally, the combination of the two names sounds like the feminine of the old Rabelaisian word: "gastrolâtre" becomes "gastrolastress" at the time when society becomes uni sexualized and when productive cycles oblige to watch one's body as women do with theirs for reproduction. The latter being in equal measure a source of stimulation and pathology that necessitates—in a reflexive relationship to one's body—a form of self-control, therapies to 'get in touch with one's inner self', ways to decompress, to let off steam through transgressive behaviours (changes in diet depending on work activity or lack of, restraint of self-control of the active body, excess linked to off-duty time). When we first conceived of this type, people born after the Second World War (who were less ritualized at the table than their elders), people in the service sector, and people living in large cities more frequently presented "gastrolastress" profiles. This has intensified, today, regarding the last two factors.

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
