# Peer review of "Reflections for a Sociological Representation of the Eater"

_socsci, doi:10.3390/socsci10090339_

Round 1

Reviewer 1 Report

The article, essentially theoretical, aims to critically review the work done by the author since the 1980s and takes up the main results of research resulting from the perspective that prioritizes the eater instead of the consumer, as was traditionally done. This reflection is based on ethno-methodological observation which is part of the current of comprehensive sociology, constructivist sociology and interactionist sociology and takes its source in classical and contemporary socio-anthropology (Mauss, Gurvitch, Mills, Duvignaud, Weber, Schutz, Berger and Luckmann, Ledrut, etc.). Taking care to distinguish sociality from sociability, the author demonstrates the effectiveness of hyper-empirical dialectical approach to find the behavioral plurality of eaters and proposes a construction of ethos of each ideal type.

This perspective, which is based on the results of empirical experiments, proposes a plural and situational approach to food processes, starting from the point of view of the actors and their practices, to update a more recent typology, the "hopping-eaters"). The author updates an explanatory model based on a typology of Eaters, foods and sharing situations (Triangles, channel and diadrama of eating), which takes into account the diachronic and synchronic dimensions and to which different ethos correspond.

This sociology of eaters that is proposed to readers has the vocation to be exhaustive and adapts to the different empirical contexts of the situations studied. In addition to the elegance, complexity and adaptability of the model, this revision aims to "reconcile the rules and dynamics specific to a particular situation, with the imprint of global society, a dimension often hidden but always present (...) and understand the upstream side of consumption, that of the values that underlie it within a consumer mentality, a consumer who is both a replicator and a producer of standards"
The article, which presents both the results of a long research commitment in the field of food and its modern components but also the way in which it is lived and thought, from the eater, shows the vitality of the model and opens up prospects for new researches, in a world that is becoming increasingly complex and where food issues are at the heart of individual and collective processes.

Author Response

Thanks for your comments. I have revised this manuscript.

Reviewer 2 Report

It is a very interesting initiative to give space to an article in English presenting these theoretical contributions to French food sociology. The following suggestions are therefore intended only to facilitate the reception of the article by the English-speaking public.

1) There are many French words, often concepts forged in that language. It is therefore legitimate to maintain the French, in the absence of a real translation. However, they should be systematically italicized and placed in inverted commas. Other French words used in the text should be italicized, as should all other non-English words.

2) The notion of "gastrolastress" forged from proposals made by François Rabelais should be made more explicit. For example, in note 8, Rabelais is probably insufficiently known to current generations of sociologists who are far from French culture. We suggest introducing a paragraph in the note to explain who this author is and what his place is in French culture. If possible, it would be good to give a reference where the notion of "gastrolastre" is presented. If, in addition, the reference is a translation into English, that would be perfect.

3 A short biographical sketch showing key dates and intellectual influences would be useful for the reader to get the measure of this article. This should be added to the presentation that the coordinator of the issue will certainly make.

4 The following references exist in English, and we suggest replacing the French version by the English one:

BERGER P., LUCKMANN T., La construction sociale de la réalité, Méridiens­ Klincksieck, Paris, 1986.

BOURDIEU P., La distinction. Critique sociale du jugement. Editions de minuit, Paris, 1979.

DUVIGNAUD J., Chebika, Gallimard 1968, Terre humaine, Plon, Paris, 1991.

KARDINER., L'Individu dans sa société : essai d'anthropologie psychanalytique, Bibliothèque des sciences humaines, éd. Gallimard, Paris, 1969.

MILLS C.W., L'imagination sociologique, Les textes à l'appui, Maspero, Paris, 1967.

SCHUTZ A., Le chercheur et le quotidien, Méridiens-Klincksieck, Paris, 1987.

WEBER M., L’éthique protestante et l’esprit du capitalisme, Plon, Paris, 1964.

5 Suggestion to replace:

DREWNOW A et SPECTER S.,   Poverty and obesity : the role of energy density and energy costs, Amer.J.Clin.Nut., 2004, (79), pp 6616.SKI

By

DREWNOWSKI A et SPECTER S.,   Poverty and obesity: the role of energy density and energy costs, Amer.J.Clin.Nut., 2004, (79), pp 6616.

Author Response

(The authors gave the same response as above.)

Reviewer 3 Report

Some English editing is needed. It is possible to notice that the article was translated from another language (probably from French) and some sentences are confused. Authors also place suspension points in sentences where they are not necessary.

In the part “The conception of ethos”, the authors present a series of theoretical ideas that should be referenced, even if it is from previous works by the authors themselves, as some of them have already been published.

Page 2 (line 74 -86). The authors present some methodological elements, but it is not clear how they will be articulated in the article. Were these methodologies used to collect the data discussed in the text?

Why do authors choose to address behavioural ethos with regard to products containing fat? What are its particularity, relevance and originality for understanding food dynamics in “food hyper-modernity”?

Page 4 (line 159-162). The authors mention that they defined, in the 1980s, four types of “eaters” and, in note 4, they present some information about the origin of these data. I suggest you indicate more clearly the characteristics of the study sample (age, gender, social class, cultural origin, etc.), as well as its methodology. For the publication of this article, these methodological aspects must be explained, also because the authors “update” their analyses and do not present any data on the methodology. Also, when was current data collected?

Page 4: What are the social characteristics of “complexed overeaters”? (Mainly in relation to social class and gender)

Page 5 (line 249): it is necessary to include a note that explains the meaning of “bobos”. Not every non-French reader will know what this term means.

At times the authors indicate that they created the « type of eaters » in the 1980s and at times in the 1990s, 2000s. It is necessary to standardize this information or to better explain this methodological aspect.

Although the authors present the translation of the diagrams below each figure, I suggest that the diagrams be entirely in English (not in French). It is also necessary to mention the reference of these diagrams, even if the same authors have published them elsewhere.

Page 9 (line 376): the authors mention a diagram 3, but this figure is not included in the manuscript.

In general, even if the structure of the article is understood, the organization is not entirely clear and deserves modifications. For example, in the introduction, the authors indicate that “We will particularly materialize this representation, which results from an empirical approach, through the example of the "gastrolastress" (emergence of an intermittent behavioral model), the true template of the "hopping-eaters" who reflect hyper-modernity" and in the last part of the work, they begin by explaining the "complexed overeaters" and presenting data that could have been indicated in part 3.1 such as their social characteristics and behaviours.

Some references that are mentioned in “references” are not cited in the text.

Through the reading of the abstract (and also keywords), we expect that the authors will make a discussion related to the concepts of “eater” and “consumer”. Although this differentiation is present indirectly, the text focuses almost exclusively on the concept of “eater”.

Author Response

(The authors gave the same response as above.)
